# Application and Diagnostic Performance of Two-Dimensional Shear Wave Elastography and Liver Fibrosis Scores in Adults with Class 3 Obesity

**DOI:** 10.3390/nu16010074

**Published:** 2023-12-25

**Authors:** Ritesh Chimoriya, Vincent Ho, Ziqi Vincent Wang, Ruby Chang, Badwi B. Boumelhem, David Simmons, Nic Kormas, Mark D. Gorrell, Milan K. Piya

**Affiliations:** 1School of Medicine, Western Sydney University, Campbelltown, NSW 2560, Australia; r.chimoriya@westernsydney.edu.au (R.C.); v.ho@westernsydney.edu.au (V.H.); da.simmons@westernsydney.edu.au (D.S.); 2Camden and Campbelltown Hospitals, Campbelltown, NSW 2560, Australia; ruby.chang@health.nsw.gov.au (R.C.); nic.kormas@health.nsw.gov.au (N.K.); 3Centenary Institute, Faculty of Medicine and Health, The University of Sydney, Sydney, NSW 2006, Australia; v.wang@centenary.org.au (Z.V.W.); b.boumelhem@centenary.org.au (B.B.B.); m.gorrell@centenary.org.au (M.D.G.)

**Keywords:** two-dimensional (2D) shear wave elastography, liver fibrosis, obesity, non-alcoholic fatty liver disease, metabolic dysfunction-associated fatty liver disease, metabolic dysfunction-associated steatotic liver disease, FIB-4, APRI, NAFLD fibrosis score, fibroblast activation protein

## Abstract

There are no ideal non-invasive tests for assessing the severity of liver fibrosis in people with metabolic dysfunction-associated steatotic liver disease (MASLD) and class 3 obesity, where body habitus often makes imaging technically challenging. This study aimed to assess the applicability and diagnostic performance of two-dimensional shear wave elastography (2D-SWE), alongside several serum-based liver fibrosis scoring methods, in individuals with class 3 obesity. A cross-sectional study was conducted in patients aged ≥18 years and with a body mass index (BMI) ≥ 40 kg/m^2^ who were participants in a publicly funded multidisciplinary weight management program in South Western Sydney. The 2D-SWE was performed using the ElastQ Imaging (EQI) procedure with the Phillips EPIQ Elite series ultrasound. An EQI Median value of ≥6.43 kPa was taken as a cutoff score for significant fibrosis, and the scan was considered valid when the liver EQI IQR/Med value was <30%. The Fibrosis-4 (FIB-4) index, AST-to-platelet ratio index (APRI), NAFLD fibrosis score (NFS), and circulating fibroblast activation protein index (FAP index) were calculated from fasting blood samples. The participants (*n* = 116; 67.2% female) were aged 47.2 ± 12.9 years, with BMI 54.5 ± 11.0 kg/m^2^. EQI Median values were obtained for 97.4% (113/116) of the 2D-SWE scans, and 91.4% (106/116) of the scans were considered valid. The EQI Median values exhibited a moderately positive correlation with the FIB-4 index (*r* = 0.438; *p* < 0.001) and a weakly positive correlation with the APRI *(r* = 0.388; *p* < 0.001), NFS (*r* = 0.210; *p* = 0.036) and FAP index (*r* = 0.226; *p* = 0.020). All liver fibrosis scores were positively correlated with one another. Among those referred for a liver biopsy based on the 2D-SWE and serum scores, half (11/22) underwent liver biopsy, and their 2D-SWE scores exhibited 72.7% accuracy (sensitivity: 71.4%; specificity: 75%) in detecting significant fibrosis. Our results show that 2D-SWE is a feasible, non-invasive test to assess liver fibrosis among people with class 3 obesity. Further research is needed to assess how 2D-SWE can be used alongside existing serum-based risk scores to reliably detect significant fibrosis, which would potentially reduce the need for invasive liver biopsy.

## 1. Introduction

Obesity, defined as a body mass index (BMI) ≥ 30 kg/m^2^ [1], is a multifactorial condition that is increasing at an alarming rate globally [2,3]. The rising prevalence of class 3 obesity (BMI ≥ 40 kg/m^2^) has also emerged as a major public health problem in several developed countries [4,5,6]. In Australia, over 31% of adults are living with obesity, and 4% are living with class 3 obesity [7]. Excess body weight is an important risk factor for mortality and morbidity from a wide range of chronic diseases, including type 2 diabetes, cardiovascular disease, and metabolic dysfunction-associated steatotic liver disease (MASLD), previously called non-alcoholic fatty liver disease (NAFLD) or metabolic dysfunction-associated fatty liver disease (MAFLD) [8,9,10,11]. Obesity is recognised as the primary risk factor for the development of MASLD [12], and the prevalence of imaging-defined MASLD has been reported to be approximately 90–95% among those with class 3 obesity [13]. Moreover, type 2 diabetes (T2DM) is also known to predict the development of MASLD, and MASLD is associated with increased prevalence of T2DM, and vice versa [14,15].

NAFLD encompasses a spectrum of liver conditions, ranging from simple hepatic steatosis to non-alcoholic steatohepatitis (NASH)/metabolic steatohepatitis (MeSH), which may progress to advanced fibrosis, cirrhosis, and hepatocellular carcinoma [16,17]. Metabolic dysfunction-associated fatty liver disease (MAFLD) was adopted as a more appropriate nomenclature than NAFLD/NASH [18]. However, the diagnostic criteria for MAFLD significantly differ from those of NAFLD, as MAFLD diagnosis is based on the presence of metabolic dysfunction, without excluding patients with alcohol intake or other chronic diseases. A recent consensus study has adopted metabolic dysfunction-associated steatotic liver disease (MASLD) as the overarching term to be used instead of NAFLD [11].

Assessment of liver fibrosis stages and monitoring the extent and progress of liver fibrosis are essential factors in the management of MASLD [19,20]. Moreover, early and accurate detection of significant liver fibrosis allows for timely management of cirrhosis and related comorbidities [21]. Conventionally, liver biopsy is recognised as the diagnostic gold standard for the staging of liver fibrosis. However, liver biopsy is an invasive procedure that risks complications, is dependent on the site of tissue sampled, and its routine use for monitoring is further limited by the technical difficulty and higher risk of complications among those with class 3 obesity [22,23,24]. Considering these limitations, several non-invasive diagnostic methods have been developed as alternatives to liver biopsy.

Non-invasive scoring systems based on routinely available clinical and laboratory parameters are particularly valuable for the initial assessment of fibrosis [25,26]. The commonly used liver fibrosis scoring systems include the Fibrosis-4 (FIB-4) index [27], aspartate aminotransferase (AST)-to-platelet ratio index (APRI) [28], and NAFLD fibrosis score (NFS) [29]. We have also previously demonstrated the potential for the use of circulating fibroblast activation protein alpha (cFAP) as a diagnostic biomarker for liver fibrosis, based on which the fibroblast activation protein (FAP) index can be calculated [30,31]. In prior studies, cFAP has been found to be elevated in cirrhosis, as well as in moderate-to-severe fibrosis among individuals with type 2 diabetes or those undergoing bariatric surgery, further demonstrating its potential clinical utility as a liver fibrosis biomarker [32,33].

Ultrasound-based elastography, which is based on the evaluation of liver stiffness measurements, is increasingly used as a non-invasive diagnostic tool to assess liver fibrosis staging [34]. Transient elastography using liver FibroScan^®^ is a pioneering ultrasound-based elastography technique that has been validated against liver histology [35]. However, higher failure rates have been observed among people with obesity, and evidence is lacking among people with class 3 obesity [36,37]. Two-dimensional shear wave elastography (2D-SWE) is a more recently developed elastography technique that has emerged as a promising alternative to liver biopsy in the assessment of liver fibrosis in MASLD, especially in individuals with obesity [37]. As 2D-SWE is integrated with a conventional ultrasound device, it allows for the assessment of liver stiffness in real time, along with simultaneous anatomic B-mode ultrasound imaging and selection of the region of interest [37,38]. However, the applicability of 2D-SWE for the assessment of liver fibrosis among those with class 3 obesity is not well established [39]. Moreover, there are no ideal non-invasive tests to assess the severity of liver fibrosis in the class 3 obese population, in whom body habitus often makes imaging technically challenging [40,41]

Therefore, this study aimed to assess the applicability and diagnostic performance of 2D-SWE, alongside several non-invasive liver fibrosis scoring systems, in individuals with class 3 obesity. The following research questions guided this study:What percentage of 2D-SWE scans were successfully conducted in people with class 3 obesity, and what proportion of these had a valid reading?How does liver fibrosis staging, as assessed by 2D-SWE, correlate with liver fibrosis scores (FIB-4 index, APRI, NAFLD fibrosis score, and FAP index) in people with class 3 obesity?How do the diagnostic performance metrics of liver fibrosis scores (FIB-4 index, APRI, NAFLD fibrosis score, and FAP index) and 2D-SWE compare to histological liver analysis in individuals who have been positively screened for MASLD and have class 3 obesity?

## 2. Methods

### 2.1. Study Design and Setting

This was a cross-sectional study conducted in a hospital-based, publicly funded, multidisciplinary weight management program in Sydney, Australia, as previously described [42,43,44,45]. The study participants were recruited between February 2021 and December 2022, and they included all newly enrolled patients in the program who attended at least one physician appointment. The included patients were over 18 years of age, with a BMI ≥ 40 kg/m^2^ and at least one weight-related medical comorbidity, most commonly MASLD or type 2 diabetes.

### 2.2. Ethics

This study was conducted in accordance with the Declaration of Helsinki, and the research conducted received approval from the South Western Sydney Local Health District Human Research Ethics Committee (Reference: 2019_ETH08677; Approval date: 23 July 2019).

### 2.3. Data Collection

Data on anthropometry, weight, medical comorbidities, and medications were collected from routine clinical data in electronic medical records and paper notes, where available. Routine blood tests were conducted at the time of enrolment in the program, which included full blood counts, HbA1c, liver function tests, kidney function tests, and lipid profiles. The liver function tests included alanine aminotransferase (ALT), albumin, alkaline phosphatase (ALP), aspartate aminotransferase (AST), bilirubin, gamma-glutamyl transferase (GGT), and total protein. Plasma samples for the assessment of cFAP activity levels were collected along with routine blood draws for other tests, centrifuged, and stored at −80 °C until used. All samples underwent analysis in one batch. FAP is stable in stored serum after several freeze–thaw rounds when stored at −80 °C long-term and at 4 °C overnight [33]. Liver stiffness measurements were obtained using 2D-SWE for all participants. Liver biopsy reports were obtained from the electronic medical records.

### 2.4. Staging of Liver Fibrosis

The degree of liver fibrosis was assessed using 2D-SWE, liver fibrosis scores (FIB-4 index, APRI, NFS, and FAP index) and, when available, histological liver analysis. Liver fibrosis was classified using the meta-analysis of histological data in viral hepatitis (METAVIR) scoring system [46], which employs a scale ranging from 0 to 4. The stages of fibrosis include F0, indicating no fibrosis, F1, representing mild fibrosis (portal fibrosis without septa), F2, indicating significant fibrosis (portal fibrosis with a few septa), F3, indicating severe fibrosis (presence of numerous septa without cirrhosis), and F4, indicating the presence of cirrhosis [46].

#### 2.4.1. Two-Dimensional (2D) Shear Wave Elastography

Two-dimensional (2D) SWE was performed using the ElastQ Imaging (EQI) procedure with the Phillips EPIQ Elite series ultrasound (Philips Healthcare, Macquarie Park, NSW, Australia). The elastography scans were performed as per the manufacturer’s manual [47]. All participants were instructed to fast for at least 6 h before the scan. During the scan, the participants were advised to be in a left lateral oblique decubitus position with the right arm in maximal extension to increase the intercostal acoustic window. The transducer was positioned in the right intercostal space, targeting liver segment VII or VIII. The region-of-interest (ROI) box was placed below the liver capsule, and patients were instructed to pause their breathing for a minimum of 4 s. Cineloops of at least six seconds were acquired, ensuring optimal sampling techniques that avoided rib shadows and blood vessels to capture at least 10 images. Reports with reliability indicators of liver EQI IQR/Med values below 30% were considered valid, as per the manufacturer’s recommendations [48]. The reporting protocol followed the METAVIR scoring system for liver fibrosis classification utilising the shear wave EQI Median values [46,47,48]. The following cutoff values were used: an EQI Median value less than 6.43 kPa suggests no/mild fibrosis (F0-1), a value between 6.43 and 9.54 kPa indicates significant fibrosis (F2), a value between 9.54 and 11.34 kPa suggests advanced fibrosis (F3), and a value higher than 11.34 kPa indicates cirrhosis (F4) [48]. Throughout the study, adherence to the safety and customisation instructions outlined in the manufacturer’s manual was maintained [47].

#### 2.4.2. FAP Index

The blood samples were anticoagulated with EDTA and centrifuged at 1400 rpm for 15 min to obtain supernatant plasma samples. The samples were stored at −80 °C until all samples were available for analysis in a single batch. The cFAP activity levels from the plasma samples were measured using an in-house enzyme activity assay, as described in prior studies [31,32,33]. The cFAP activity was categorised into three ordinal levels: level 0 (cFAP activity ≤ 730 U/L), level 1 (730 U/L < cFAP activity < 1580 U/L), and level 2 (cFAP activity ≥ 1580 U/L). The cFAP activity was assigned to the score calculation according to its respective ordinal level. The score was derived from age, diabetes status, ALT, and cFAP level, as described previously [30] and as shown in Table 1.

#### 2.4.3. Liver Fibrosis Assessment Scores

Based on routine blood tests and clinical data, the FIB-4 index for liver fibrosis was calculated using Sterling’s formula [27], the APRI was calculated using Wai’s formula [28], and the NFS was calculated using Angulo’s formula [29]. Table 1 includes the scoring equations and the cutoff values for significant/advanced fibrosis used in this study.

#### 2.4.4. Histological Liver Analysis

Following the completion of all blood tests and elastography scans, a physician in the program reviewed the results of all participants. If the liver fibrosis scores (FIB-4 and/or APRI) or 2D-SWE value (EQI Median) exceeded the cutoff values for significant fibrosis, the participants were referred to a gastroenterologist for the evaluation of liver fibrosis, including a liver biopsy where indicated, as part of the assessment. A percutaneous liver biopsy was performed in the right lobe of the liver under the guidance of ultrasound or computerised tomography (CT), with a needle calibre of 16 gauge. The obtained biopsy samples consisted of 2–4 cores of grey tissue measuring 1 mm in diameter and more than 20 mm in length. These biopsy samples were subsequently fixed in formalin, consolidated into a single paraffin block, and examined by an experienced liver pathologist.

### 2.5. Data Analysis

The results are presented as means with their corresponding 95% confidence intervals. The data were analysed using the Statistical Package for the Social Sciences, Version 29 (SPSS for MacOS, SPSS Inc., Chicago, IL, USA). The normality of continuous variables was assessed using the Kolmogorov–Smirnov test. Parametric data were analysed using independent *t*-tests, and non-parametric data were analysed using the Mann–Whitney U test. Categorical variables were analysed using the chi-squared test, and the results were reported as frequencies and percentages. Pearson’s correlation coefficient was computed to assess the strength and direction of linear relationships between all scores. Sensitivity, specificity, positive predictive value, negative predictive value, positive likelihood ratio, negative likelihood ratio, and accuracy were calculated for the liver fibrosis scores (FIB-4 index, APRI, NAFLD fibrosis score, and FAP index) and 2D-SWE. These metrics were compared to the histological findings from the liver biopsies, which served as the primary comparators for diagnostic procedures.

## 3. Results

### 3.1. Participant Characteristics

A total of 116 participants enrolled in the multidisciplinary weight management program between February 2021 and December 2022 were included in this study. The baseline characteristics of the study participants are presented in Table 2. In brief, 67.2% of the participants were female, and the average age of the participants was 47.2 ± 12.9 years. The participants had an average weight of 153.6 ± 33.4 kg and an average BMI of 54.5 ± 11.0 kg/m^2^, with 28.5% having a BMI ≥ 60 kg/m^2^.

### 3.2. Application of 2D Shear Wave Elastography in Class 3 Obesity

The liver EQI Median values were obtained for 97.4% (113/116) of the 2D-SWE scans, with 2.6% (3/116) of the scans experiencing technical issues and failures to capture at least 10 images. Among the 113 successful scans, 6.2% (7/113) had a reliability indicator (liver EQI IQR/Med) above 30% and were considered to be invalid. In total, 91.4% (106/116) of the patients had a valid scan performed and were included in the final analysis. There were no statistically significant differences in weight, BMI, or waist circumference between patients who had valid scans and those who did not.

### 3.3. Non-Invasive Tools for Screening for Liver Fibrosis in Class 3 Obesity

The distribution of scores across different non-invasive screening tools varied in proportion to the number of individuals positioned on either side of the cutoff values for significant fibrosis (Figure 1). With the NFS, more than half of the participants were above the cutoff, and the proportion was even higher in people with type 2 diabetes, as BMI and diabetes status were part of the scoring system.

The EQI Median was not significantly different between people with and without type 2 diabetes (Table 3). However, the FIB-4 index showed a significant difference (*p* = 0.016) between people with and without type 2 diabetes. Similarly, the NFS and the FAP index differed significantly when diabetes was present, as both of these tools include the presence of diabetes as a key component of their score calculations.

### 3.4. Correlations between Non-Invasive Liver Fibrosis Screening Tools in Class 3 Obesity

The EQI Median had a moderate positive correlation with the FIB-4 index (*r* = 0.438; *p* < 0.001) and a weak positive correlation with the APRI *(r* = 0.388; *p* < 0.001), NFS (*r* = 0.210; *p* = 0.036), and FAP index (*r* = 0.226; *p* = 0.020) (Figure 2). The FIB-4 index had a strong positive correlation with the APRI (*r* = 0.667, *p* < 0.001), NFS (*r* = 0.603, *p* < 0.001), and FAP index (*r* = 0.616, *p* < 0.001) (Appendix A). Moreover, the APRI exhibited a weak positive correlation with the NFS (*r* = 0.222, *p* = 0.021) and FAP index (*r* = 0.320, *p* < 0.001). Similarly, the NFS had a moderate positive correlation with the FAP index (*r* = 0.547, *p* < 0.001).

### 3.5. Findings from Histological Liver Analysis

Of the 22 participants referred for liver biopsy, 11 participants underwent the procedure (50%). Histological findings revealed the presence of significant/advanced fibrosis (F ≥ 2) in 63.6% (7/11) of the participants who underwent the biopsy. The diagnostic performance metrics of 2D-SWE and the liver fibrosis scores are shown in Table 4 and Appendix A. The accuracy rates for the 2D-SWE, FIB-4 index, APRI, and FAP index were 72.7%, 81.8%, 63.6%, and 63.6%, respectively. The NFS was positive for all cases, which precluded calculation of the corresponding metrics.

## 4. Discussion

The results of this study indicate that 2D-SWE is a feasible, non-invasive technique for assessing liver fibrosis in individuals with class 3 obesity, with valid scans successfully performed for most of the participants (91.4%; 106/116). The 2D-SWE-derived EQI Median exhibited moderate-to-weak positive correlations with several established non-invasive liver fibrosis scoring systems, including the FIB-4 index, APRI, NFS, and FAP index. The diagnostic accuracy of 2D-SWE and the FIB-4 index (71.7% and 81.8%, respectively) in detecting significant fibrosis, derived from the participants who underwent liver biopsy (11/22 referred), suggests their potential utility as effective initial screening tools among individuals with class 3 obesity.

We found that 2D-SWE had a high success rate among individuals with class 3 obesity (mean BMI 54.5 ± 11.0 kg/m^2^), which underscores its applicability in this population. Despite the technical challenges posed by increased subcutaneous adipose tissue, EQI Median values were obtained for 97.4% (113/116) of the 2D-SWE scans, with 91.4% (106/116) considered to be valid scans. This finding is consistent with previous research conducted in Iran among individuals with severe obesity (mean BMI 45.35 ± 6.16 kg/m^2^), where the success rate of 2D-SWE was 97.3% (108/111 patients) [39]. A high success rate of 2D-SWE has often been reported in patients with obesity [37], which could be attributed to the use of B-mode ultrasound imaging in 2D-SWE that allows for optimisation of the probe position to obtain successful readings [39]. Conversely, higher failure rates have been noted among people with obesity and severe obesity for other imaging-based techniques, including the widely used and validated transient elastography [52,53]. In a prior Australian study among individuals with severe obesity (BMI ≥ 35 kg/m^2^; mean BMI 45.1 ± 8.3 kg/m^2^), the success rates of transient elastography in obtaining valid measurements with controlled attenuation parameters and magnetic resonance spectroscopy were only 80.5% and 65.3%, respectively [54]. Overall, the promising findings in relation to the high technical success rate of 2D-SWE strongly support its feasibility for the assessment of liver fibrosis in individuals with class 3 obesity.

We found moderate-to-weak positive correlations between 2D-SWE-derived EQI Median and several established non-invasive liver fibrosis scoring systems, most notably the FIB-4 index. This is in line with the findings of a prior Australian study, in which 2D-SWE-measured liver stiffness measurements were found to be positively correlated with the FIB-4 index and APRI [41]. Similar to the findings of this study, a prior study conducted in Korea also reported that liver stiffness measured using 2D-SWE showed a moderate and weak positive correlation with the FIB-4 index (*r* = 0.493) and APRI (*r* = 0.392), respectively [55]. Moreover, all of the liver fibrosis scores in the current study had a weak-to-moderate positive correlation with one another. Overall, our findings suggest a potential complementary role of 2D-SWE alongside these non-invasive scoring systems, and a combination of imaging and blood-based, non-invasive tests may be useful for the initial assessment of liver fibrosis among individuals with class 3 obesity [56]. Moreover, as 2D-SWE is integrated into a conventional ultrasound system, enabling simultaneous ultrasound examination and liver stiffness measurements, there is also the possibility of easily incorporating 2D-SWE into routine ultrasound procedures that are regularly performed among individuals with class 3 obesity [37].

The determination of the diagnostic accuracy of the non-invasive liver fibrosis screening tools in this study was based on a small subset of participants who underwent liver biopsy (*n* = 11/22 referred). Among the non-invasive tests, 2D-SWE and the FIB-4 index showed higher accuracy rates (71.7% and 81.8%, respectively) in identifying significant fibrosis, signifying their potential utility as effective initial screening tools among individuals with class 3 obesity. A previous study among patients with severe obesity also noted good accuracy of 2D-SWE for liver fibrosis grading, with area under the receiver operating curve (AUROC) values of 0.77, 0.72, 0.77, and 0.70 for ≥F1, ≥F2, ≥F3, and F4, respectively [39]. Conversely, a prior systematic review and meta-analysis reported high diagnostic accuracy of 2D-SWE for staging fibrosis in adults with NAFLD [57]. Among the non-invasive tests, the highest mean AUROC values for predicting significant fibrosis were reported for 2D-SWE (0.89; range 0.85–0.92), followed by other tests, including the FIB-4 index (0.75), NFS (0.72), and APRI (0.70). However, the NFS yielded positive results for all cases in the present study, and as a result, its diagnostic performance could not be compared with individuals who underwent liver biopsy. Moreover, a large proportion of the individuals included in our study had NFS scores exceeding the cutoff value (57.5%; *n* = 61/106), in contrast to the other non-invasive screening tools, where only 4.7% to 25.5% had scores above the respective cutoff values for significant fibrosis. These findings suggest that, in individuals with class 3 obesity, NFS may not be an effective screening tool, considering the large proportion of false positives generated by the current cutoff value. Nonetheless, a recent study suggested that the standard cutoff values of the NFS should be adjusted upward to prevent high rates of false positive classifications in individuals with a high BMI, as the NFS score calculation includes BMI as a component [51]. Therefore, a separate BMI-adjusted cutoff for the NFS may be required for individuals with class 3 obesity [51]. While we did not see a correlation between BMI and the FAP index, a previous study found a significant positive association between BMI and cFAP activity levels in people with normal BMI [31]. Future research could explore the complex relationship between cFAP activity levels and body weight among people with obesity.

In the current study, there were no significant differences in the 2D-SWE-derived EQI Median between individuals with and without type 2 diabetes. The significant differences observed in the FIB-4 index and NAFLD score between the two groups were likely due to age differences between the groups, as well as the presence of diabetes as a key component in the score calculations. This finding is particularly important, as research suggests that there exists a strong association between class 3 obesity and type 2 diabetes [44]. Moreover, MASLD and type 2 diabetes are known to frequently coexist, with each condition driving the progression of the other [15]. Nonetheless, large-scale studies are needed to investigate the association between the progression of MASLD and the presence of type 2 diabetes among individuals with class 3 obesity. However, these findings suggest the potential use of 2D-SWE in combination with other non-invasive tests for early screening of liver fibrosis among people with both class 3 obesity and type 2 diabetes, as well as for monitoring patients with MASLD and type 2 diabetes and their responses to treatment to mitigate the risk of adverse clinical outcomes [37,58].

A strength of this study is its real-world clinical setting and inclusion of patients with class 3 obesity enrolled in a hospital-based, publicly funded, multidisciplinary weight management program. This study successfully addressed a significant gap in the literature by investigating the applicability of 2D-SWE, a novel non-invasive diagnostic tool, in individuals with class 3 obesity, a population known to be at elevated risk of MASLD, type 2 diabetes, and associated complications [13,59]. Another strength of this study is the inclusion of several established non-invasive liver fibrosis scoring systems, alongside 2D-SWE, which allowed for comprehensive comparisons involving both imaging-based and blood-based non-invasive tests. Furthermore, the inclusion of a small subset of participants who also underwent liver biopsy provided a valuable benchmark for assessing the diagnostic accuracy of these non-invasive techniques compared to the gold-standard diagnostic method among individuals with class 3 obesity.

A limitation of this study is the relatively small subset of participants who underwent liver biopsy (*n* = 11/22 referred) for assessing the diagnostic accuracy of the non-invasive tests. This limitation arose because this study was conducted in a real-life clinical setting, where non-invasive tests are employed to limit the need for biopsy, such that the invasive liver biopsy procedure was only recommended when the liver fibrosis scores (FIB-4 and/or APRI) or 2D-SWE (EQI Median) exceeded the cutoff values for significant fibrosis, such that false negatives could not be identified. Another limitation is that this was a single-centre study conducted in a specialist clinic that required a referral for participation. Considering these limitations, large-scale validation studies encompassing diverse populations are warranted to ascertain whether 2D-SWE can be used to reliably detect significant fibrosis in individuals with class 3 obesity. Future research should further explore the potential integration of 2D-SWE in combination or for sequential use with the blood-based liver fibrosis scoring systems to improve the clinical utility of non-invasive tests, potentially limiting the need for invasive liver biopsy. Long-term longitudinal studies would be valuable to assess the effectiveness of these non-invasive techniques for monitoring the progression of liver fibrosis in individuals with class 3 obesity over time. These would also be useful to monitor liver fibrosis during weight loss interventions such as those with GLP1 agonists or bariatric surgery, as well as in investigational studies for newer therapeutic antifibrotic agents.

## 5. Conclusions

The results of this study demonstrate that 2D-SWE is a feasible, non-invasive method for assessing liver fibrosis in individuals with class 3 obesity, with valid scans successfully performed for the majority of the participants. Moderate-to-weak positive correlations were observed between 2D-SWE-derived EQI Median and established non-invasive liver fibrosis scoring systems, including the FIB-4 index, APRI, NFS, and FAP index. Among the non-invasive tests, 2D-SWE and the FIB-4 index showed higher accuracy rates in identifying significant fibrosis in participants who underwent liver biopsy. This suggests their potential utility as initial screening tools for this high-risk population, in combination or sequentially, and as tools to monitor changes in liver fibrosis over time or following specific interventions. Nonetheless, further research is warranted, with a larger sample size and validation against liver biopsy as well as non-invasive liver fibrosis risk scoring systems, to assess whether 2D-SWE can be used to reliably detect significant fibrosis, potentially reducing the need for invasive liver biopsy.

## Figures and Tables

**Figure 1 nutrients-16-00074-f001:**
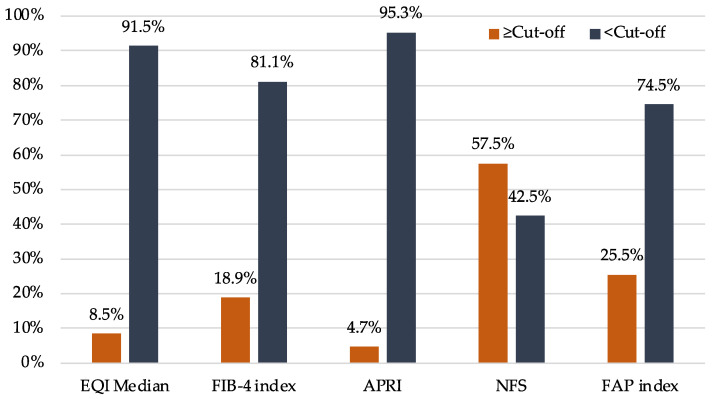
Distribution of scores across different non-invasive screening tools to assess liver fibrosis. APRI: AST (aspartate aminotransferase)-to-platelet ratio index; EQI: ElastQ Imaging; FAP: fibroblast activation protein; FIB-4: Fibrosis-4; NFS: NAFLD (non-alcoholic fatty liver disease) fibrosis score.

**Figure 2 nutrients-16-00074-f002:**
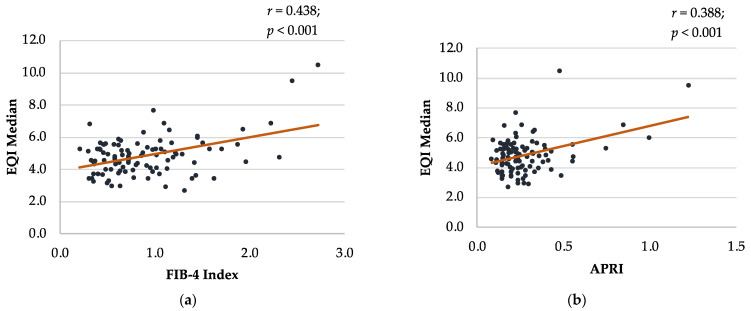
Bland–Altman plot comparing EQI Median with other non-invasive liver fibrosis screening tools in class 3 obesity. (**a**) EQI Median and FIB-4 Index; (**b**) EQI Median and APRI; (**c**) EQI Median and NFS; (**d**) EQI Median and FAP Index. APRI: AST (aspartate aminotransferase)-to-platelet ratio index; EQI: ElastQ Imaging; FAP: fibroblast activation protein; FIB-4: Fibrosis-4; NFS: NAFLD (non-alcoholic fatty liver disease) fibrosis score.

**Table 1 nutrients-16-00074-t001:** Scoring equations for liver fibrosis assessment scores and cutoff values used in the study for detecting the risk of significant/advanced fibrosis (F ≥ 2).

Scores	Equation	Cutoff
FIB-4 index [27]	[Age × AST]/[Platelet count × √ALT]	1.3 [49]
APRI [28]	[(AST/upper limit of normal)/Platelet count] × 100	0.7 [50]
NFS [29]	−1.675 + [0.037 × Age] + [0.094 × BMI] + [1.13 × IFG/diabetes (yes = 1, no = 0)] + [0.99 × AST/ALT] − [0.013 × Platelet count] − [0.66 × Albumin]	0.544 [51]
FAP index [30]	−9.499 + [0.101 × Age] + [1.533 × Diabetes (yes = 1, no = 0)] + [0.009 × ALT] + [1.158 × cFAP level (0/1/2)]	−1.681

Abbreviations: ALT: alanine aminotransferase; AST: aspartate aminotransferase; APRI: AST-to-platelet ratio index; BMI: body mass index; cFAP: circulating fibroblast activation protein alpha; FAP: fibroblast activation protein; FIB-4: Fibrosis-4; IFG: impaired fasting glucose; NFS: NAFLD (non-alcoholic fatty liver disease) fibrosis score. Units: age: years; albumin: g/dl; ALT: U/L; AST: U/L; BMI: kg/m^2^; platelet count: multiplied by 109/L. Note: the upper limit of AST considered in the equation was 40 U/L.

**Table 2 nutrients-16-00074-t002:** Baseline characteristics of the study participants.

Variable Mean ± SD or *n* (%)	All Participants (*n* = 116)	With T2DM (*n* = 42)	Without T2DM (*n* = 74)	*p*-Value
*Sociodemographic Characteristics*				
Age (years)	47.2 ± 12.9	52.4 ± 10.3	44.3 ± 13.4	<0.001 *
Female	78 (67.2%)	26 (61.9%)	52 (70.3%)	0.356
Caucasian ethnicity	79 (68.1%)	25 (59.5%)	54 (73.0%)	0.135
In paid employment	34 (29.3%)	6 (14.3%)	28 (37.8%)	0.007 *
*Anthropometry and Comorbidities*				
Weight (kg)	153.6 ± 33.4	150.5 ± 34.1	155.3 ± 33.2	0.464
BMI (kg/m^2^)	54.5 ± 11.0	54.8 ± 10.7	54.4 ± 11.3	0.853
Waist circumference	142.9 ± 17.1	146.3 ± 15.1	141.4 ± 17.8	0.203
Hypertension	62 (53.4%)	27 (64.3%)	35 (47.3%)	0.078
Dyslipidaemia	54 (46.6%)	27 (64.3%)	27 (36.5%)	0.004 *
Cardiovascular disease	16 (13.8%)	9 (21.4%)	7 (9.5%)	0.072
Obstructive sleep apnoea	63 (54.3%)	25 (59.5%)	38 (51.4%)	0.396
GORD	44 (37.9%)	22 (52.4%)	22 (29.7%)	0.016 *
Thyroid disorder	12 (10.3%)	3 (7.1%)	9 (12.2%)	0.394

* Significant at *p* < 0.05. BMI: body mass index. GORD: gastro-oesophageal reflux disease. T2DM: type 2 diabetes.

**Table 3 nutrients-16-00074-t003:** Assessment of liver fibrosis in the study participants using non-invasive screening tools.

Variable Mean (95% CI)	with T2DM (*n* = 34)	without T2DM (*n* = 72)	*p*-Value
*Liver Enzymes*			
ALT (IU/L)	35.3 (26.5, 44.1)	36.1 (31.1, 41.0)	0.392
AST (IU/L)	28.9 (21.6, 36.2)	25.8 (22.8, 28.8)	0.759
GGT (IU/L)	54.2 (40.2, 68.2)	44.1 (36.9, 51.4)	0.311
*Two-Dimensional (2D) Shear Wave Elastography*			
EQI Median (kPa)	5.0 (4.5, 5.6)	4.8 (4.5, 5.0)	0.697
*Liver Fibrosis Assessment Scores*			
FIB-4 index	1.0 (0.8, 1.2)	0.8 (0.7, 0.9)	0.016
APRI	0.3 (0.2, 0.4)	0.3 (0.2, 0.3)	0.672
NFS	1.5 (1.0, 1.9)	0.4 (0.0, 0.8)	<0.001
FAP index	−1.2 (−1.6, −0.8)	−3.0 (−3.4, −2.7)	<0.001

ALT: alanine aminotransferase; AST: aspartate aminotransferase; APRI: AST-to-platelet ratio index; EQI: ElastQ Imaging; FAP: fibroblast activation protein; FIB-4: Fibrosis-4; GGT: gamma-glutamyl transferase; NFS: NAFLD (non-alcoholic fatty liver disease) fibrosis score; T2DM: type 2 diabetes.

**Table 4 nutrients-16-00074-t004:** Diagnostic performance metrics of 2D shear wave elastography and liver fibrosis scores in NAFLD-positive participants.

Screening Tool	Sensitivity	Specificity	PPV	NPV	PLR	NLR	Accuracy
2D-SWE	71.4%	75%	83.3%	60%	2.9	0.4	72.7%
FIB-4 index	100%	50%	77.8%	100%	2.0	0	81.8%
APRI	42.9%	100%	100%	50%	-	0.6	63.6%
FAP index	75.0%	33.3%	75.0%	33.3%	1.1	0.8	63.6%
Combination of 2D-SWE and FIB-4 index	88.8%	50%	88.8%	50%	1.8	0.2	81.8%
Combination of 2D-SWE and APRI	62.5%	66.6%	83.3%	40%	1.8	0.6	63.6%
Combination of 2D-SWE and FAP index	87.5%	33.3%	77.8%	50%	1.3	0.4	72.7%
Combination of 2D-SWE, FIB-4 index, APRI, and FAP index	100%	33.3%	80%	100%	1.5	0	81.8%

2D-SWE: 2D shear wave elastography; APRI: AST (aspartate aminotransferase)-to-platelet ratio index; FAP: fibroblast activation protein; FIB-4: Fibrosis-4; NLR: negative likelihood ratio; NPV: negative predictive value; PLR: positive likelihood ratio; PPV: positive predictive value.

## Data Availability

The data presented in this study are available on request from the corresponding author. The data are not publicly available due to privacy.

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
