# Peer review of "Application and Diagnostic Performance of Two-Dimensional Shear Wave Elastography and Liver Fibrosis Scores in Adults with Class 3 Obesity"

_nutrients, 2023, doi:10.3390/nu16010074_

Round 1
Reviewer 1 Report
Comments and Suggestions for Authors
Review: SWE in Obesity Class 3
This paper explores the results of SWE in very obese people, where the average BMI was 54.
It shows that it is feasible and may be helpful, although the strength of a simple test like the FIB-4 is again documented. In summary at least 91% of very obese patients can undergo a successful measurement of liver elasticity. Moreover, the PPV for stages F2-F4 was best (table 4). It follows that because has a NPV of 100% (measured against liver biopsy), it could be used first, and the SWE done when FIB-4 exceeds 1.3.
The paper should be slimmed down.
Please see my comments below:
1. Consider change NAFLD to MASLD throughout, per recent nomenclature changes (Hepatology November 2023)
2. Page 5. More data on biopsy are needed. What was the gauge of the biopsy? Its total length? I would not accept the fibrosis score in a biopsy smaller than 20 mm and with a gauge greater that 16G. Even though the number of biopsies was only 11, the data could be very valuable.
3. Fig 1 should be deleted: it does not really bring anything to the article
4. Fig 2: Typically an R value, in correlative statistics, is considered positive if R > 0.5. Even a R=0.6 would mean that R2 being 0.36, only 36% of 1 value can be inferred from the other. Thus, a R~ 0.4 is quite weak in predicting a correlation. This being said, the best R value is between FIB-4 and SWE. Indeed table 4 reinforces the well known NPV of FIB-4.
5. Fig 3 shows well known correlations between several NIT’s. The value of cFAP is very limited (see table 4), at least in this specific population. It should be deleted.
6. Paragraph from line 331-346 should be reworded in view of my comments on R values and their interpretation.
7. Lines 377-381. The FIB-4 is greater in diabetics simply because they are older (Table 2) and this drives FIB-4 higher.
8. Page 15 (Conclusions). The real strength of the paper is the description of SWE in this very specific class of very obese patients and it’s worth reporting on. I would tone down the collelation aspect and emphasize the usefulness of sequential FIB-4 and SWE approach to select patients for therapy (GLP-1 agonists or surgery) or for investigational studies.
Author Response
We appreciate your time and expertise in thoroughly reviewing our manuscript, please find attached responses and revised manuscript.

Reviewer 2 Report
Comments and Suggestions for Authors
In this manuscript, the authors evaluated the applicability of 2D-SWE as a non-invasive technique to assess liver fibrosis in patients with class 3 obesity.
It is a well thought out study project and necessary controls to evaluate the success of 2D-SWE has been considered.
The authors have also recognized that it cannot be a standalone technique, but rather be supplemented along with other currently applied assays.
Author Response

(The authors gave the same response as above.)
